# Weightless: lossy weight encoding for deep neural network compression

## Abstract

The large memory requirements of deep neural networks limit their deployment and adoption on many devices. Model compression methods effectively reduce the memory requirements of these models, usually through applying transformations such as weight pruning or quantization. In this paper, we present a novel scheme for lossy weight encoding which complements conventional compression techniques. The encoding is based on the Bloomier filter, a probabilistic data structure that can save space at the cost of introducing random errors. Leveraging the ability of neural networks to tolerate these imperfections and by re-training around the errors, the proposed technique, Weightless, can compress DNN weights by up to $496\times$ with the same model accuracy. This results in up to a $1.51\times$ improvement over the state-of-the-art.

## 1 Introduction

The continued success of deep neural networks (DNNs) comes with increasing demands on compute, memory, and networking resources. Moreover, the correlation between model size and accuracy suggests that tomorrow's networks will only grow larger. This growth presents a challenge for resource-constrained platforms such as mobile phones and wireless sensors. As new hardware now enables executing DNN inferences on these devices (Apple, 2017; Qualcomm, 2017), a practical issue that remains is reducing the burden of distributing the latest models especially in regions of the world not using high-bandwidth networks. For instance, it is estimated that, globally, 800 million users will be using 2G networks by 2020 (GSMA, 2014), which can take up to 30 minutes to download just 20MB of data. By contrast, today's DNNs are on the order of tens to hundreds of MBs, making them difficult to distribute. In addition to network bandwidth, storage capacity on resource-constrained devices is limited, as more applications look to leverage DNNs. Thus, in order to support state-of-the-art deep learning methods on edge devices, methods to reduce the size of DNN models without sacrificing model accuracy are needed.

Model compression is a popular solution for this problem. A variety of compression algorithms have been proposed in recent years and many exploit the intrinsic redundancy in model weights. Broadly speaking, the majority of this work has focused on ways of simplifying or eliminating weight values (e.g., through weight pruning and quantization), while comparatively little effort has been spent on devising techniques for encoding and compressing.

In this paper we propose a novel lossy encoding method, *Weightless*, based on Bloomier filters, a probabilistic data structure (Chazelle et al., 2004). Bloomier filters inexactly store a function map, and by adjusting the filter parameters, we can elect to use less storage space at the cost of an increasing chance of erroneous values. We use this data structure to compactly encode the weights of a neural network, exploiting redundancy in the weights to tolerate some errors. In conjunction with existing weight simplification techniques, namely pruning and clustering, our approach dramatically reduces the memory and bandwidth requirements of DNNs for over the wire transmission and on-device storage. Weightless demonstrates compression rates of up to $496\times$ without loss of accuracy, improving on the state of the art by up to $1.51\times$. Furthermore, we show that Weightless scales better with increasing sparsity, which means more sophisticated pruning methods yield even more benefits.

This work demonstrates the efficacy of compressing DNNs with *lossy* encoding using probabilistic data structures. Even after the same aggressive lossy simplification steps of weight pruning and clustering (see Section 2), there is still sufficient extraneous information left in model weights to allow

an approximate encoding scheme to substantially reduce the memory footprint without loss of model accuracy. Section 3 reviews Bloomier filters and details Weightless. State-of-the-art compression results using Weightless are presented in Section 4. Finally, in Section 4.3 shows that Weightless scales better as networks become more sparse compared to the previous best solution.

## 2 RELATED WORK

Our goal is to minimize the static memory footprint of a neural network without compromising accuracy. Deep neural network weights exhibit ample redundancy, and a wide variety of techniques have been proposed to exploit this attribute. We group these techniques into two categories: (1) methods that modify the loss function or structure of a network to reduce free parameters and (2) methods that compress a given network by removing unnecessary information.

The first class of methods aim to directly train a network with a small memory footprint by introducing specialized structure or loss. Examples of specialized structure include low-rank, structured matrices of Sindhwani et al. (2015) and randomly-tied weights of Chen et al. (2015). Examples of specialized loss include teacher-student training for knowledge distillation (Bucila et al., 2006; Hinton et al., 2015) and diversity-density penalties (Wang et al., 2017). These methods can achieve significant space savings, but also typically require modification of the network structure and full retraining of the parameters.

An alternative approach, which is the focus of this work, is to compress an existing, trained model. This exploits the fact that most neural networks contain far more information than is necessary for accurate inference (Denil et al., 2013). This extraneous information can be removed to save memory. Much prior work has explored this opportunity, generally by applying a two-step process of first *simplifying* weight matrices and then *encoding* them in a more compact form.

Simplification changes the number or characteristic of weight values to reduce the information needed to represent them. For example, pruning by selectively zeroing weight values (LeCun et al., 1989; Guo et al., 2016) can, in some cases, eliminate over 99% of the values without penalty. Similarly, most models do not need many bits of information to represent each weight. Quantization collapses weights to a smaller set of unique values, for instance via reduction to fixed-point binary representations (Gupta et al., 2015) or clustering techniques (Gong et al., 2014).

Simplifying weight matrices can further enable the use of more compact encoding schemes, improving compression. For example, two recent works Han et al. (2016); Choi et al. (2017) encode pruned and quantized DNNs with sparse matrix representations. In both works, however, the encoding step is a lossless transformation, applied on top of lossy simplification.

## 3 WEIGHTLESS

Weightless is a lossy encoding scheme based around Bloomier filters. We begin by describing what a Bloomier filter is, how to construct one, and how to retrieve values from it. We then show how we encode neural network weights using this data structure and propose a set of augmentations to make it an effective compression strategy for deep neural networks.

### 3.1 THE BLOOMIER FILTER

A Bloomier filter generalizes the idea of a Bloom filter (Bloom, 1970), which are data structures that answer queries about set membership. Given a subset $S$ of a universe $U$, a Bloom filter answers queries of the form, "Is $v \in S$?". If $v$ is in $S$, the answer is always yes; if $v$ is not in $S$, there is some probability of a false positive, which depends on the size of the filter, as size is proportional to encoding strength. By allowing false positives, Bloom filters can dramatically reduce the space needed to represent the set. A Bloomier filter (Chazelle et al., 2004) is a similar data structure but instead encodes a function. For each $v$ in a domain $S$, the function has an associated value $f(v)$ in the range $R = [0, 2^r)$. Given an input $v$, a Bloomier filter always returns $f(v)$ when $v$ is in $S$. When $v$ is not in $S$, the Bloomier filter returns a null value $\perp$, except that some fraction of the time there is a "false positive", and the Bloomier filter returns an incorrect, non-null value in the range $R$.

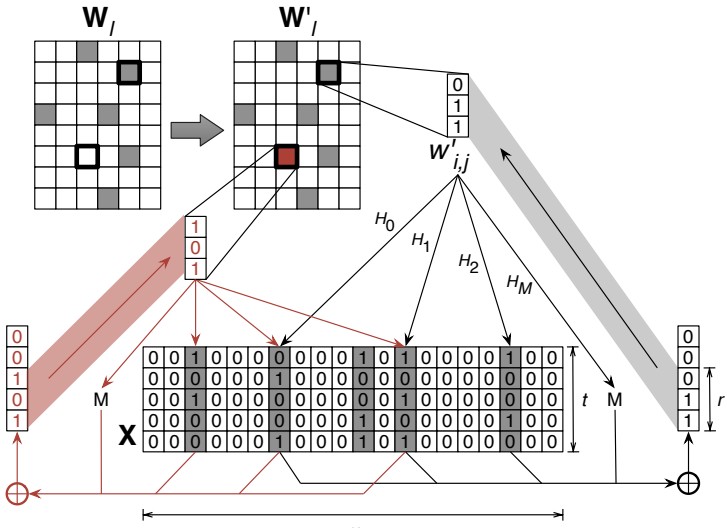

Figure 1: Encoding with a Bloomier filter. $\mathbf{W}'$ is an inexact reconstruction of $\mathbf{W}$ from a compressed projection $\mathbf{X}$. To retrieve the value $w'_{i,j}$, we hash its location and exclusive-or the corresponding entries of $\mathbf{X}$ together with a computed mask $M$. If the resulting value falls within the range $[0, 2^{r-1})$, it is used for $w'_{i,j}$, otherwise, it is treated as a zero. The red path on the left shows a false positive due to collisions in $\mathbf{X}$ and random $M$ value.

**Decoding** Let $S$ be the subset of values in $U$ to store, with $|S| = n$. A Bloomier filter uses a small number of hash functions (typically four), and a hash table $\mathbf{X}$ of $m = cn$ cells for some constant $c$ (1.25 in this paper), each holding $t > r$ bits. For hash functions $H_0, H_1, H_2, H_M$, let $H_{0,1,2}(v) \rightarrow [0, m)$ and $H_M(v) \rightarrow [0, 2^r)$, for any $v \in U$. The table $\mathbf{X}$ is set up such that for every $v \in S$,

$$X_{H_0(v)} \oplus X_{H_1(v)} \oplus X_{H_2(v)} \oplus H_M(v) = f(v).$$

Hence, to find the value of $f(v)$, hash $v$ four times, perform three table lookups, and exclusive-or together the four values. Like the Bloom filter, querying a Bloomier filter runs in $O(1)$ time. For $u \notin S$, the result, $X_{H_0(u)} \oplus X_{H_1(u)} \oplus X_{H_2(u)} \oplus H_M(u)$, will be uniform over all $t$-bit values. If this result is not in $[0, 2^r)$, then $\perp$ is returned and if it happens to land in $[0, 2^r)$, a false positive occurs and a result is (incorrectly) returned. An incorrect value is therefore returned with probability $2^{r-t}$.

**Encoding** Constructing a Bloomier filter involves finding values for $\mathbf{X}$ such that the relationship above holds for all values in $S$. There is no known efficient way to do so directly. All published approaches involve searching for configurations with randomized algorithms. In their paper introducing Bloomier filters, Chazelle et al. (2004) give a greedy algorithm which takes $O(n \log n)$ time and produces a table of size $\lceil cn \rceil t$ bits with high probability. Charles & Chellapilla (2008) provide two slightly better constructions. First, they give a method with identical space requirements but runs in $O(n)$ time. They also show a separate $O(n \log n)$-time algorithm for producing a smaller table with $c$ closer to 1. Using a more sophisticated algorithm for construction should allow for a more compact table and, by extension, improve the overall compression rate. However, we leave this for future work and use the method of (Chazelle et al., 2004) for simplicity.

While construction can be expensive, it is a one-time cost. Moreover, the absolute runtime is small compared to the time it takes to train a deep neural network. In the case of VGG-16, our unoptimized Python code built a Bloomier filter within an hour. We see this as a small worthwhile overhead given the savings offered and in contrast to the days it can take to fully train a network.

### 3.2 APPROXIMATE WEIGHT ENCODING WITH BLOOMIER FILTERS

We propose using the Bloomier filter to compactly store weights in a neural network. The function $f$ encodes the mapping between indices of nonzero weights to their corresponding values. Given a

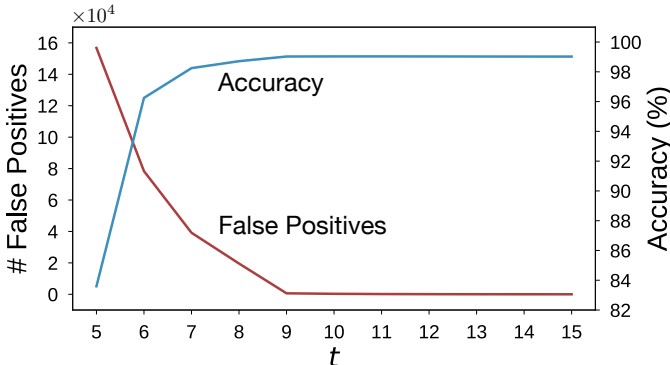

Figure 2: There is an exponential relationship between $t$ and the number of false positives (red) as well as the measured model accuracy with the incurred errors (blue).

weight matrix $\mathbf{W}$, define the domain $S$ to be the set of indices $\{i, j \mid w_{i,j} \neq 0\}$. Likewise, the range $R$ is $[-2^{a-1}, 2^{a-1}) - \{0\}$ for $a$ such that all values of $\mathbf{W}$ fall within the interval. Due to weight value clustering (see below) this range is remapped to $[0, 2^{r-1})$ and encodes the cluster indices. A null response from the filter means the weight has a value of zero.

Once $f$ is encoded in a filter, an approximation $\mathbf{W}'$ of the original weight matrix is reconstructed by querying it with all indices. The original nonzero elements of $\mathbf{W}$ are preserved in the approximation, as are most of the zero elements. A small subset of zero-valued weights in $\mathbf{W}'$ will take on nonzero values as a result of random collisions in $\mathbf{X}$, possibly changing the model's output. Figure 1 illustrates the operation of this scheme: An original nonzero is correctly recalled from the filter on the right and a false positive is created by an erroneous match on the left (red).

**Complementing Bloomier filters with simplification** Because the space used by a Bloomier filter is $O(nt)$, they are especially useful under two conditions: (1) The stored function is sparse (small $n$, with respect to $|U|$) and (2) It has a restricted range of output values (small $r$, since $t > r$). To improve overall compression, we pair approximate encoding with weight transformations.

Pruning networks to enforce sparsity (condition 1) has been studied extensively (Hassibi & Stork, 1993; LeCun et al., 1989). In this paper, we consider two different pruning techniques: (i) magnitude threshold plus retraining and (ii) dynamic network surgery (DNS) (Guo et al., 2016). Magnitude pruning with retraining as straightforward to use and offers good results. DNS is a recently proposed technique that prunes the network during training. We were able to acquire two sets of models, LeNet-300-100 and LeNet5, that were pruned using DNS and include them in our evaluation; as no reference was published for VGG-16 only magnitude pruning is used. Regardless of how it is accomplished, improving sparsity will reduce the overall encoding size linearly with the number of non-zeros with no effect on the false positive rate (which depends only on $r$ and $t$). The reason for using two methods is to demonstrate the benefits of Weightless as networks increase in sparsity, the DNS networks are notably more sparse than the same networks using magnitude pruning.

Reducing $r$ (condition 2) amounts to restricting the range of the stored function or minimizing the number of bits required to represent weight values. Though many solutions to discretize weights exist (e.g., limited binary precision and advanced quantization techniques Choi et al. (2017)), we use $k$-means clustering. After clustering the weight values, the $k$ centroids are saved into an auxiliary table and the elements of $\mathbf{W}$ are replaced with indices into this table. This style of indirect encoding is especially well-suited to Bloomier filters, as these indices represent a small, contiguous set of integers. Another benefit of using Bloomier filters is that $k$ does not have to be a power of 2. When decoding Bloomier filters, the result of the XORs can be checked with an inequality, rather than a bitmask. This allows Bloomier filters to use $k$ exactly, reducing the false positive rate by a factor of $1 - \frac{k}{2^r}$. In other methods, like that of Han et al. (2016), there is no benefit, as any $k$ not equal to a power of two strictly wastes space.

**Tuning the $t$ hyperparameter** The use of Bloomier filters introduces an additional hyperparameter $t$, the number of bits per cell in the Bloomier filter. $t$ trades off the Bloomier filter's size and the false positive rate which, in turn, effects model accuracy. While $t$ needs to be tuned, we find it far easier to

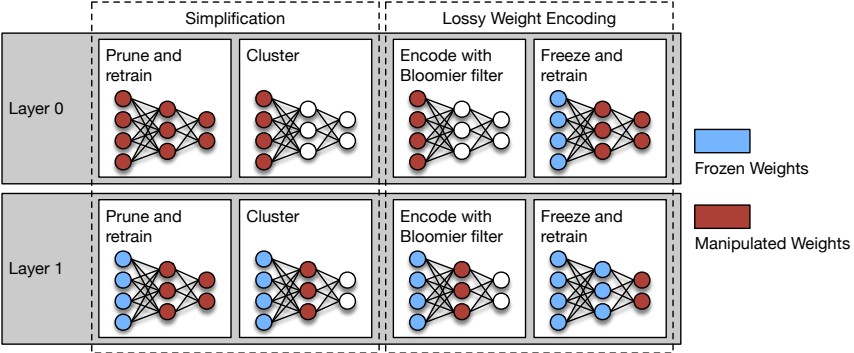

Figure 3: Weightless operates layer-by-layer, alternating between simplification and lossy encoding. Once a Bloomier filter is constructed for a weight matrix, that layer is frozen and the subsequent layers are briefly retrained (only a few epochs are needed).

reason about than other DNN hyperparameters. Because we encode $k$ clusters, $t$ must be greater than $\lceil \log_2 k \rceil$, and each additional $t$ bit reduces the number of false positives by a factor of 2. This limits the number of reasonable values for $t$: when $t$ is too low the networks experience substantial accuracy loss, but also do not benefit from high values of $t$ because they have enough implicit resilience to handle low error rates (see Figure 2). Experimentally we find that $t$ typically falls in the range of 6 to 9 for our models.

**Retraining to mitigate the effects of false positives** We encode each layer's weights sequentially. Because the weights are fixed, the Bloomier filter's false positives are deterministic. This allows for the retraining of deeper network layers to compensate for errors. It is important to note that encoded layers are not retrained. The randomness of the Bloomier filter would sacrifice all benefits of mitigating the effects of errors. If the encoded layer was retrained, a new encoding would have to be constructed (because $S$ changes) and the indices of weights that result in false positives would differ after every iteration of retraining. Instead, we find retraining all subsequent layers to be effective, typically allowing us to reduce $t$ by one or two bits. The process of retraining around faults is relatively cheap, requiring on the order of tens of epochs to converge. The entire optimization pipeline is shown in Figure 3.

**Compressing Bloomier filters** When sending weight matrices over a network or saving them to disk, it is not necessary to retain the ability to access weight values as they are being sent, so it is advantageous to add another layer of compression for transmission. We use arithmetic coding, an entropy-optimal stream code which exploits the distribution of values in the table (MacKay, 2005). Because the nonzero entries in a Bloomier filter are, by design, uniformly distributed values in $[1, 2^t - 1)$, improvements from this stage largely come from the prevalence of zero entries.

## 4 EXPERIMENTS

We evaluate Weightless on three deep neural networks commonly used to study compression: LeNet-300-100, LeNet5 (LeCun et al., 1998), and VGG-16 (Simonyan & Zisserman, 2015). The LeNet networks use MNIST (Lecun & Cortes) and VGG-16 uses ImageNet (Russakovsky et al., 2015). The networks are trained and tested in Keras (Chollet, 2017). The Bloomier filter was implemented in house and uses a Mersenne Twister pseudorandom number generator for uniform hash functions. To reduce the cost of constructing the filters for VGG-16, we shard the non-zero weights into ten separate filters, which are built in parallel to reduce construction time. Sharding does not significantly affect compression or false positive rate as long as the number of shards is small (Broder & Mitzenmacher, 2004).

We focus on applying Weightless to the largest layers in each model, as shown in Table 1. This corresponds to the first two fully-connected layers of LeNet-300-100 and VGG-16. For LeNet5, the second convolutional layer and the first fully-connected layer are the largest. These layers account for 99.6%, 99.7%, and 86% of the weights in LeNet5, LeNet-300-100, and VGG-16, respectively.

Table 1: **Baseline.** Summary of accuracy, baseline parameters (sparsity and number of clusters), and Weightless' hyperparameter ($t$) setting used for each layer. *VGG-16 in MB (size) and top-1 (error).

| Model | | | Baseline | | | | |
| --- | --- | --- | --- | --- | --- | --- | --- |
| | Pruning Method | Error % | Layer | Size (KB) | Nonzero % | Clusters | $t$ |
| LeNet-300-100 | Magnitude | 1.76 | FC-0 | 919 | 5.0 | 9 | 8 |
| | | | FC-1 | 117 | 5.0 | 9 | 9 |
| | DNS | 2.03 | FC-0 | 919 | 1.8 | 9 | 9 |
| | | | FC-1 | 117 | 1.8 | 10 | 8 |
| LeNet5 | Magnitude | 0.98 | CNN-1 | 36 | 7.0 | 9 | 8 |
| | | | FC-0 | 2304 | 5.5 | 9 | 7 |
| | DNS | 0.96 | CNN-1 | 98 | 3.1 | 10 | 8 |
| | | | FC-0 | 1564 | 0.73 | 10 | 8 |
| VGG-16* | Magnitude | 35.9 | FC-0 | 392 | 2.99 | 4 | 6 |
| | | | FC-1 | 64 | 4.16 | 4 | 8 |

(The DNS version is slightly different than magnitude pruning, however the trend is the same.) While other layers could be compressed, they offer diminishing returns.

**Compression baseline** The results below present both the absolute compression ratio and the improvements Weightless achieves relative to Deep Compression (Han et al., 2016), which represents the current state-of-the-art. The absolute compression ratio is with respect the original standard 32-bit weight size of the models. This baseline is commonly used to report results in publications and, while larger than many models used in practice, it provides the complete picture for readers to draw their own conclusions. For comparison to a more aggressive baseline, we reimplemented Deep Compression in Keras. Deep Compression implements a lossless optimization pipeline where pruned and clustered weights are encoded using compressed sparse row encoding (CSR) and then compresses CSR encoding tables with Huffman coding. The compression achieved by Deep Compression we use as a baseline is notably better than the original publication (e.g., VGG-16 FC-0 went from $91\times$ to $119\times$).

**Error baseline** Because Weightless performs lossy compression, it is important to bound the impact of the loss. We establish this bound as the error of the trained network after the simplification steps (i.e., post pruning and clustering). In doing so, the test errors from compressing with Weightless and Deep Compression are the same (shown as Baseline Error % in Table 1). Weightless is sometimes slightly better due to training fluctuations, but never worse. While Weightless does provide a tradeoff between compression and model accuracy, we do not consider it here. Instead, we feel the strongest case for this method is to compare against a lossless technique with iso-accuracy and note compression ratio will only improve in any use case where degradation in model accuracy is tolerable.

## 4.1 SPARSE WEIGHT ENCODING

Given a simplified baseline model, we first evaluate the how well Bloomier filters encode sparse weights. Results for Bloomier encoding are presented in Table 2, and show that the filters work exceptionally well. In the extreme case, the large fully connected layer in LeNet5 is compressed by $445\times$. With encoding alone and demonstrates a $1.99\times$ improvement over CSR, the alternative encoding strategy used in Deep Compression.

Recall that the size of a Bloomier filter is proportional to $mt$, and so sparsity and clustering determine how compact they can be. Our results suggest that sparsity is more important than the number of clusters for reducing the encoding filter size. This can be seen by comparing each LeNet models' magnitude pruning results to those of dynamic network surgery—while DNS needs additional clusters, the increased sparsity results in a substantial size reduction. We suspect this is due to the ability of DNNs to tolerate a high false positive rate. The $t$ value used here is already on the exponential part of the false positive curve (see Figure 2). At this point, even if $k$ could be reduced, it is unlikely $t$ can be since the additional encoding strength saved by reducing $k$ does little to protect against the

Table 2: **Lossy encoding.** Weight matrices encoded using Bloomier filters (Weightless) are smaller than those encoded with CSR (Deep Compression), without loss of accuracy. In addition, Weightless tends to do relatively better on larger models and when using more advanced pruning algorithms. The Improvement column shows Bloomier filters are up to $1.99\times$ more efficient than CSR.

| Model | Pruning Method | Layer | Compression Factor (Size KB) | | |
| | | | CSR | Weightless | Improvement |
|---|---|---|---|---|---|
| LeNet-300-100 | Magnitude | FC-0 | $40.1\times$ (22.9) | $40.6\times$ (20.1) | $1.01\times$ |
| | | FC-1 | $46.9\times$ (2.50) | $56.1\times$ (2.09) | $1.20\times$ |
| | DNS | FC-0 | $112\times$ (8.22) | $152\times$ (6.04) | $1.36\times$ |
| | | FC-1 | $99.0\times$ (1.18) | $174\times$ (0.67) | $1.75\times$ |
| LeNet5 | Magnitude | CNN-1 | $40.7\times$ (0.89) | $46.2\times$ (0.78) | $1.14\times$ |
| | | FC-0 | $46.6\times$ (46.6) | $66.6\times$ (34.6) | $1.43\times$ |
| | DNS | CNN-1 | $80.6\times$ (1.21) | $97.8\times$ (1.00) | $1.21\times$ |
| | | FC-0 | $224\times$ (6.99) | $445\times$ (3.52) | $1.99\times$ |
| VGG-16 | Magnitude | FC-0 | $81.8\times$ (4790) | $142\times$ (2750) | $1.74\times$ |
| | | FC-1 | $71.2\times$ (900) | $74.6\times$ (860) | $1.05\times$ |

doubling of false positives when in this range. For VGG-16 FC-0, there are more false positives in the reconstructed weight matrix than there are non-zero weights originally; using $t = 6$ results in over 6.2 million false positives while after simplification there are only 3.07 million weights. Before recovered with retraining, Bloomier filter encoding increased the top-1 error by 2.0 percentage points. This is why we see Bloomier filters work so well here–most applications cannot function with this level of approximation, nor do they have an analogous retrain mechanism to mitigate the errors' effects.

Table 3: **Network compression.** Encoded weights can be compressed further for transmission or storage. Below are the results of applying arithmetic coding to Bloomier filters and Huffman coding to CSR. The Improvement column shows Weightless offers up to a $1.51\times$ improvement over Deep Compression.

| Model | Pruning Method | Layer | Compression Factor (Size KB) | | |
| | | | Huffman | Weightless | Improvement |
|---|---|---|---|---|---|
| LeNet-300-100 | Magnitude | FC-0 | $59.1\times$ (15.6) | $60.1\times$ (15.3) | $1.02\times$ |
| | | FC-1 | $56.0\times$ (2.09) | $64.3\times$ (1.82) | $1.15\times$ |
| | DNS | FC-0 | $153\times$ (5.98) | $174\times$ (5.27) | $1.13\times$ |
| | | FC-1 | $129\times$ (0.91) | $195\times$ (0.60) | $1.51\times$ |
| LeNet5 | Magnitude | CNN-1 | $42.8\times$ (0.84) | $51.6\times$ (0.70) | $1.21\times$ |
| | | FC-0 | $59.1\times$ (39.0) | $74.2\times$ (31.1) | $1.25\times$ |
| | DNS | CNN-1 | $89.5\times$ (1.09) | $114.4\times$ (0.86) | $1.28\times$ |
| | | FC-0 | $333\times$ (4.70) | $496\times$ (3.16) | $1.49\times$ |
| VGG-16 | Magnitude | FC-0 | $119\times$ (3280) | $157\times$ (2500) | $1.31\times$ |
| | | FC-1 | $88.4\times$ (720) | $85.8\times$ (740) | $0.97\times$ |

## 4.2 COMPRESSING WEIGHT ENCODINGS

For sending a model over a network, an additional stage of compression can be used to optimize for size. Han et al. (2016) propose using Huffman coding for their, and we use arithmetic coding, as described in Section 3.2. The results in Table 3 show that while Deep Compression gets relatively more benefit from a final compression stage, Weightless remains a substantially better scheme overall. Prior work by Mitzenmacher (2002) on regular Bloom filters has shown that they can be optimized

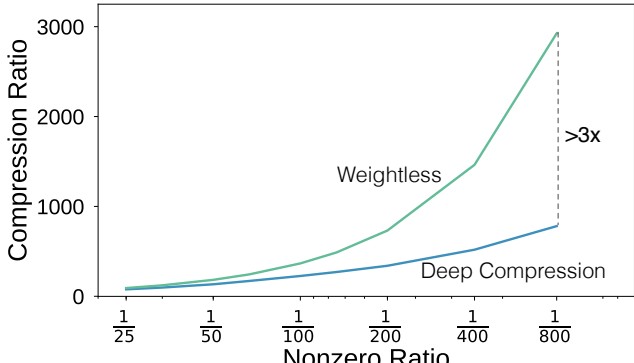

Figure 4: Weightless exploits sparsity more effectively than Deep Compression. By setting pruning thresholds to produce specific nonzero ratios, we can quantify sparsity scaling. There is no loss of accuracy at any point in this plot.

for better post-compression size. We believe a similar method could be used on Bloomier filters, but we leave this for future work.

### 4.3 SCALING WITH SPARSITY

Recent work continues to demonstrate better ways to extract sparsity from DNNs (Guo et al., 2016; Ullrich et al., 2017; Narang et al., 2017), so it is useful to quantify how different encoding techniques scale with increasing sparsity. As a proxy for improved pruning techniques, we set the threshold for magnitude pruning to produce varying ratios of nonzero values for LeNet5 FC-0. We then perform retraining and clustering as usual and compare encoding with Weightless and Deep Compression (all without loss of accuracy). Figure 4 shows that as sparsity increases, Weightless delivers far better compression ratios. Because the false positive rate of Bloomier filters is controlled independent of the number of nonzero entries and addresses are hashed not stored, Weightless tends to scale very well with sparsity. On the other hand, as the total number of entries in CSR decreases, the magnitude of every index grows slightly, offsetting some of the benefits.

### 5 CONCLUSION

This paper demonstrates a novel lossy encoding scheme, called Weightless, for compressing sparse weights in deep neural networks. The lossy property of Weightless stems from its use of the Bloomier filter, a probabilistic data structure for approximately encoding functions. By first simplifying a model with weight pruning and clustering, we transform its weights to best align with the properties of the Bloomier filter to maximize compression. Combined, Weightless achieves compression of up to $496\times$, improving the previous state-of-the-art by $1.51\times$.

We also see avenues for continuing this line of research. First, as better mechanisms for pruning model weights are discovered, end-to-end compression with Weightless will improve commensurately. Second, the theory community has already developed more advanced—albeit more complicated— construction algorithms for Bloomier filters, which promise asymptotically better space utilization compared to the method used in this paper. Finally, by demonstrating the opportunity for using lossy encoding schemes for model compression, we hope we have opened the door for more research on encoding algorithms and novel uses of probabilistic data structures.

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

## 6 APPENDIX

### 6.1 CONSTRUCTING BLOOMIER FILTERS

Given a set of key value pairs, in this case weight addresses and cluster indexes, a Bloomier filter is constructed as follows. The task of construction is to find a unique location in the table for each key such that the value can be encoded there. For each to be encoded key a *neighborhood* of 3 hash digests are generated, indexes of these 3 are named iotas.

To begin, the neighborhoods of all keys are computed and the unique digests are identified. Keys with unique neighbors are removed from the list of keys to be encoded. When a key is associated with a unique location, its iota value (i.e., the unique neighbor index into the neighborhood of hashs), is saved in an ordered list along with the key itself. The process is repeated for the remaining keys and continues until either all keys identify a unique location or none can be found. In the case that this search fails a different hash function can be tried or a larger table is required (increasing $m$).

Once the unique locations and ordering is known, the encoding can begin. Values are encoded into the filter in the reverse order in which the unique locations are found during the search described above. This is done such that as non-unique neighbors of keys collide, they can still resolve the correct encoded values. For each key, the unique neighbor (indexed by the saved iota value) is set as the XOR of all neighbor filter entries (each a $t$-bit length vector), a random mask $M$, and the key's value. As the earlier found unique key locations are populated in the filter, it is likely that the neighbor values will be non-zero. By XORing together them in reverse order the encoding scheme guarantees that the correct value is retrieved. (The XORs can be thought of cancelling out all the erroneous information except that of the true value.)

An implementation of the Bloomier filter will be released along with publication.

### 6.2 SUPPLEMENTAL MATERIAL

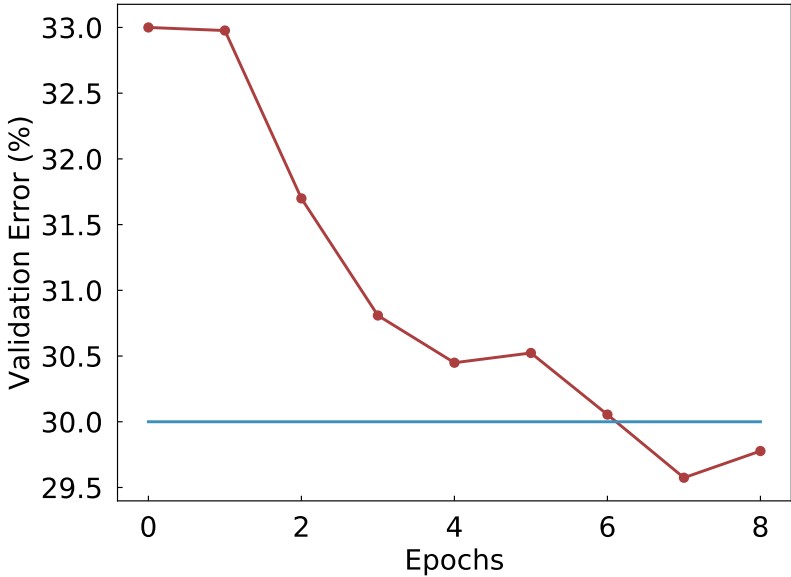

Figure 5: Plot of retraining (as described in Section 3.2) VGG16's FC-1 after encoding FC-0 in a Bloomier filter. The baseline validation accuracy (on 25000 samples) is 30% and after encoding FC-0, with a Bloomier filter, this increases to 33%. As the plots hows, after a few epochs pass the error introduced by the lossy encoding is recovered. In comparison, test accuracy after encoding FC-0 and before retraining FC-1 is 38.0% and goes down by 2% to 36.0% which maintains the model's baseline performance.

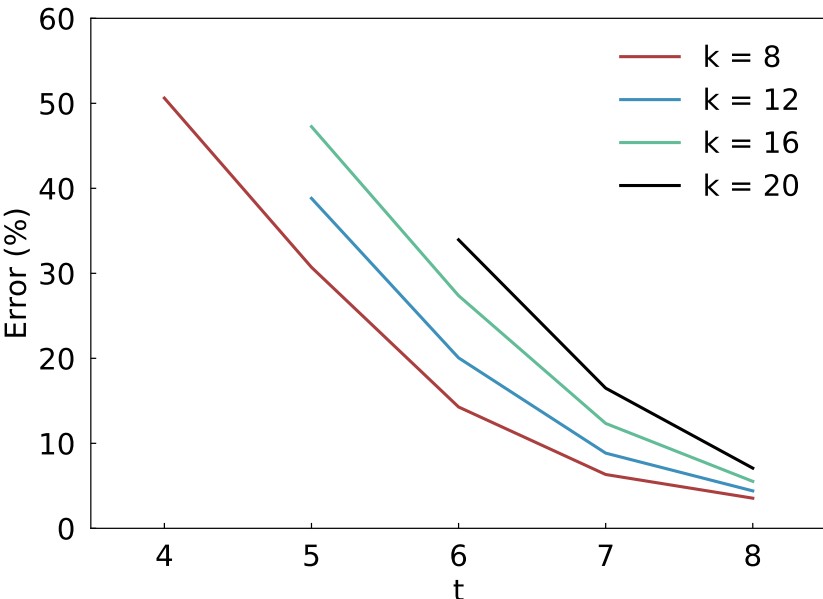

Figure 6: Sensitivity analysis of $t$ with respect to the number of weight clusters, $k$, for FC-0 in LeNet-300-100. As per Figure 2, model error decreases with the number of false positives, which are determined by the difference between $t$ and $k$. This is emphasized here as $k = 8$ performs strictly better than higher $k$ values as it results in the strongest encoding given $t$.

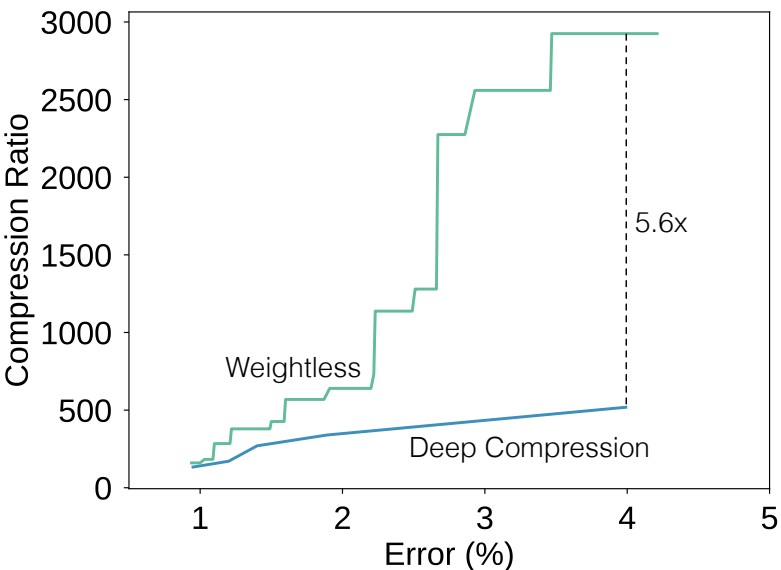

Figure 7: Demonstrating iso-accuracy comparisons between previous state-of-the-art DNN compression techniques and Weightless for FC-0 in LeNet5. In comparison to Table 3, we see that with respect to accuracy, Weightless' compression ratio scales significantly better than Deep Compression. This is advantageous in applications that are amenable to reasonable performance loss. To generate this plot we swept the pruning threshold for both Weightless and Deep Compression, and $t$ from 6 to 9 for Weightless.

