# OpenReview forum: "Weightless: Lossy Weight Encoding For Deep Neural Network Compression"
_ICLR.cc/2018/Conference — Invite to Workshop Track_

### Official Review · AnonReviewer3 · 2017-11-26
**ICLR 2018 review. Weightless**

**Rating:** 6
**Confidence:** 4

**Review:**

Summary: The paper addresses the actual problem of compression of deep neural networks. Authors propose to use another technique for sparse matrix storage. Namely, authors propose to use Bloomier filter for more efficient storage of sparse matrices obtained from Dynamic Network Surgery (DNS) method. Moreover, authors propose elegant and efficient trick for mitigating errors of Bloomier filter. Overall, the paper present clear and completed research of the proposed technique.
Clarity and Quality: The paper is well structured and easy to follow. Authors provide a reader with a large amount of technical details, which helps to reproduce their method. The paper contains detailed investigation of every step and aspect in the proposed pipeline, which made this research well-organized and complete.
Though the presentation of some results can be improved. Namely, core values for compression and improvement are presented only for two biggest layers in networks, but more important values are compression and improvement for whole networks.
Originality and Significance: The main contribution of the paper is the adaptation of Bloomier filter for sparse network obtained from almost any procedure of networks sparsification. However, this adaptation is almost straightforward, except the proposed trick of network fine-tuning for compensating false positive values of Bloomier filter. Significance of the results is hard to estimate because of several reasons:
Values of compression and improvement are presented only for two layers, not for the whole network.
According to Fig. 4, encoding of sparse matrices via Bloomier filter is efficient (compared to CSR) only for matrices with nonzero ratio greater than 0.04. So this method can’t be applied to all layers in network, that can significantly influence overall compression.

Other Comments:
The procedure of network evaluation is totally omitted in the paper. So a model supposed to be “unpacked” (to the dense or CSR format) before evaluation. Considering this, comparison with CSR could be made only for sending the model over a network. Since, CSR format can be efficiently used during evaluation.
Minor Comments:
(Page 5) “Because we encode $k$ clusters, $t$ must be greater than $\lceil \log_2 k\rceil$”. Perhaps, “...$r$ must be greater than $\lceil \log_2 k\rceil$” would be better for understanding.
(Page 7) “Bloomier filter encoding increased the top-1 accuracy by 2.0 percentage points”. Perhaps, authors have meant top-1 error.

---

> ### Author Response · Authors · 2017-12-21
> **Response to Reviewer3**
>
> "Values of compression and improvement are presented only for two layers, not for the whole network.
> According to Fig. 4, encoding of sparse matrices via Bloomier filter is efficient (compared to CSR) only for matrices with nonzero ratio greater than 0.04. So this method can’t be applied to all layers in network, that can significantly influence overall compression."
>
> Figure 4 is meant to demonstrate that Weightless scales better with increasing sparsity than Deep Compression. The results we present in Table 2 show that Weightless does not require a non-zero ratio of 4%; CNN-1 in LeNet5 with magnitude pruning has a non-zero ratio of 7% and it still outperforms Deep Compression.
>
>
> "The procedure of network evaluation is totally omitted in the paper. So a model supposed to be “unpacked” (to the dense or CSR format) before evaluation. Considering this, comparison with CSR could be made only for sending the model over a network. Since, CSR format can be efficiently used during evaluation."
>
> You are correct. In its current form, a model would need to be “unpacked” to the dense format before evaluation, but this paper is meant to focus solely on compression for over the wire transmission. We feel this is an important problem facing companies deploying deep learning models. We are currently investigating building specialized hardware for efficient sparse processing that would enable evaluating models in the encoded space.

---

### Official Review · AnonReviewer2 · 2017-11-26
**Lacking evaluations with few technical concerns**

**Rating:** 6
**Confidence:** 4

**Review:**

The problem of lossy compression of neural networks is essentially important and relevant. The paper proposes an interesting usage of Bloomier filters in lossy compression of neural net weights. The Bloomier filter is proposed by others. It is a data structure that maps from sparse indices to their corresponding values with chances that returns incorrect values for non-existing indices. The paper compares its method with two baseline methods (Magnitude and Dynamic network surgery DNS) to demonstrates its performance.

I find the paper fairly interesting but still have some concerns in the technical part and experiments.

Pros:
1. The paper seems the first to introduce Bloomier filter into the network compression problem. I think its contribution is novel and original. The paper may interest those who work in the network compression domain.
2. The method works well in the demonstrated experimental cases.

Cons:
1. The technical part is partially clear. It might be worthwhile to briefly describe the encoding/construction algorithm used in the paper. It is recommended to describe a bit more details about how such encoding/decoding methods are applied in reducing neural net weights.
2. One drawback of the proposed method is that it has to work with sparse weights. That requires the method to be used together with network pruning methods, which seems limiting its applicability. I believe the paper can be further improved by including a study of the compression results without a pruning method (e.g., comparing with Huffman in table 3).
3. What is the reason there is no DNS results reported for VGG-16? Is it because the network is deeper?
4. The experimental part can be improved by reporting the compression results for the whole network instead of a single layer.
5. It seems the construction of Bloomier filter is costly and the proposed method has to construct Bloomier filters for all layers. What is the total time cost in terms of encoding and decoding those networks (LeNet and VGG)? It would be nice to have a separate comparison on the time consumption of  different methods.
6. Figure 4 seems a bit misleading. The comparison should be conducted on the same accuracy level instead of the ratio of nonzero weights. I recommend producing another new figure of doing such comparison.
7. The proposed idea seems somewhat related to using low rank factorization of weight matrices for compression. It might be worthwhile to compare the two approaches in experiments.
8. I am specifically interested in discussions about the possibility of encoding the whole network instead of layer-by-layer retraining.

---

> ### Author Response · Authors · 2017-12-21
> **Response to Reviewer2**
>
> 1. The technical part is partially clear. It might be worthwhile to briefly describe the encoding/construction algorithm used in the paper. It is recommended to describe a bit more details about how such encoding/decoding methods are applied in reducing neural net weights.
>
> We have included a brief description of construction in the appendix. Also, as now mentioned in the paper, we will release an implementation with the publication of the paper.
>
> 2. One drawback of the proposed method is that it has to work with sparse weights. That requires the method to be used together with network pruning methods, which seems limiting its applicability. I believe the paper can be further improved by including a study of the compression results without a pruning method (e.g., comparing with Huffman in table 3).
>
> You are correct that sparsity is necessary. This is also true for competing encoding techniques (namely Deep Compression). For the large VGG16 fully connected layer we ran Huffman encoding on un-pruned, clustered weights and got an 12.8x compression factor, which is an order of magnitude less than the reported results.
>
> 3. What is the reason there is no DNS results reported for VGG-16? Is it because the network is deeper?
>
> No, it was because the weights were not made available and were unclear how to tune the DNS hyperparameters to effectively prune the VGG16 weights. We would like to include a DNS version of VGG16 as the suspected improvement in sparsity would likely significantly improve our results. We are actively working on more advanced pruning techniques, different model types, and datasets to demonstrate the benefits of lossy encoding. We feel this paper presents the core technique and benefits of the proposed method over the state-of-the-art.
>
> 4. The experimental part can be improved by reporting the compression results for the whole network instead of a single layer.
>
> We focused on the largest layers to get the most benefit. In the final version we can report the overall compression if the reviewers feel it is beneficial.
>
> 5. It seems the construction of Bloomier filter is costly and the proposed method has to construct Bloomier filters for all layers. What is the total time cost in terms of encoding and decoding those networks (LeNet and VGG)? It would be nice to have a separate comparison on the time consumption of  different methods.
>
> Construction times for LeNet300-100, LeNet5, and VGG16 are 6 seconds, 23 seconds, and 517 seconds, respectively. Decoding takes 11, 12, and 505 seconds for each of the aforementioned models. We believe that these one-time overheads are negligible considering the significant reductions in model size.
>
> 6. Figure 4 seems a bit misleading. The comparison should be conducted on the same accuracy level instead of the ratio of nonzero weights. I recommend producing another new figure of doing such comparison.
>
> Thank you for the suggestion. We believe that this suggestion can improve the paper. As a result, we conducted additional experiments on iso-accuracy comparison between Weightless and Deep Compression in Figure 7 (appendix).
>
> 7. The proposed idea seems somewhat related to using low rank factorization of weight matrices for compression. It might be worthwhile to compare the two approaches in experiments.
>
> We believe the benefits of low rank factorization lie in efficient execution by reducing the number of computations requires. A byproduct of low rank factorization is compression on the order of 50%. However, when specifically targeting over the wire compression, this is not competitive with existing techniques. If you feel that this is an important distinction that must be made, we will happily add it in the related work section.
>
> 8. I am specifically interested in discussions about the possibility of encoding the whole network instead of layer-by-layer retraining.
>
> We can encode the whole network by eliminating the retraining steps. However, this will come at the expense of either model accuracy (if we use the same t value as with retraining) or overall compression (if we increase t). For example, without retraining, VGG16 can lose 2% absolute accuracy as shown in  Figure 5 (appendix). Previously, we tried using an auxiliary data structure to fix false positives (called exception lists), this proved to incur significant storage overheads.  As a result, we strongly believe that retraining is an integral part of mitigating the effects of false positives.

---

### Official Review · AnonReviewer1 · 2017-11-27
**Interesting idea, limited applicability (storing and transmission of models), very limited results and missing details. I would like to see clearer and more comprehensive results in terms of modern models and in the complete model, not only in the FC layer, including accuracy impact.**

**Rating:** 4
**Confidence:** 4

**Review:**

This paper proposes an interesting approach to compress the weights of a network for storage or transmission purposes. My understanding is, at inference, the network is 'recovered' therefore there is no difference in processing time (slight differences in accuracy due to the approximation in recovering the weights).

- The idea is nice although it's applicability is limited as it is only for distribution of the model and storing (is storage really a problem?).

Method:
- the idea of using the Bloomier filter is new to me. However, the paper is miss-leading as the filtering is a minor part of the complete process. The paper introduces a complete pipeline including quantization, and pruning to maximize the benefits of the filter and an additional (optional) step to achieve further compression.

- The method / idea seems simply and easy to reproduce (except the subsequent steps that are not clearly detailed).

Clarity

- The paper could improve its clarity. At the moment, the Bloomier is the core but needs many other components to make it effective. Those components are not detailed to the level of being reproducible.
- One interesting point is the self-implementation of the Deep compression algorithm. The paper claims this is a competitive representation as it achieves better compression than the original one. However, those numbers are not clear in tables (only in table 3 numbers seem to be equivalent to the ones in the text). This needs clarification, CSR achieves 81.8% according to Table 2 and 119 according to the text.

Results:
- Current results are interesting. However I have several concerns:
1) it is not clear to me why assuming similar performance. While Bloomier is weightless the complete process involves many retraining steps involving performance loss. Analysis on this would be nice to see (I doubt it ends exactly at the same number). Section 3 explicitly suggest there is the need of retraining to mitigate the effect of false positives which is then increased with pruning and quantization. Therefore, would be nice to see the impact in accuracy (even it is not the main focus of the work).

2) Resutls are focused on fully connected layers which carry (for the given models) the larger number of weights (and therefore it is easy to get large compression numbers). What would happen in newer models where the fully connected layer is minimal compared to conv. layers? What about the accuracy impact there? Let's say in a Resnet-34.
3) I would like to see further analysis on why Bloomier filter encoding improves accuracy (or is a typo and meant to be error?) by 2%. This is a large improvement without training from scractch.
4) It is interesting to me how the retraining process is 'hidden' all over the paper. At the beginning it is claimed that it takes about one hour for VGG-16 to compute the Bloomier filters. Howerver, that is only a minimal portion of the entire pipeline. Later in the experimental section it is mentioned that 'tens of epochs' are needed for retraining (assuming to compensate for errors) after retraining for compensating l1 pruning?.... tens of epochs is a significant portion of the entire training process assuming VGG is trained for 90epochs max.

5) Interestingly, as mentioned in the paper, this is 'static compression'. That is, the model needs to be completely 'restored' before inference. This is miss-leading as an embedded device will need the same requirements as any other at inferece time(or maybe I am missing something). That is, the benefit is mainly for storing and transmission.

6) I would like to see the sensibility analysis with respect to t and the number of clusters.

7) As mentioned before, LeNet is great but would be nice to see more complicated models (even resnet on CIFAR). These models are not only large in terms of parameters but also quite sensitive to modifications in the weight structure.

8) Results are focused on a single layer. What happens if all the layers are considered at the same time? Here I am also concerned about the retraining process (fixing one layer and retraining the deeper ones). How is this done using only fully connected layers? What is the impact of doing it all over the network (let's say VGG-16 from the first convolutional layer to the very last).

Summary:

All in all, the idea has potential but there are many missing details. I would like to see clearer and more comprehensive results in terms of modern models and in the complete model, not only in the FC layer, including accuracy impact.

---

> ### Author Response · Authors · 2017-12-21
> **Response to Reviewer 1**
>
> "The paper could improve its clarity..."
> The reason for the brevity on pruning and clustering was because we viewed these aspects as prior work and did not want to spend time discussing materials we did not deem research contributions of our manuscript.
>
> "One interesting point is the self-implementation of the Deep compression algorithm....This needs clarification, CSR achieves 81.8% according to Table 2 and 119 according to the text."
> We understand your confusion and will clarify the text, but these numbers are indeed correct. The 81.8x is encoding only (using CSR) and the 119x is CSR+Huffman. The distinction is there to compare Bloomier filters with Bloomier+LZMA (i.e., encoding in 4.1 with compression in 4.2).
>
> 1) "It is not clear to me why assuming similar performance..."
> This is an excellent point and to address it we have added a plot (see Figure 5) to the appendix of the paper that shows how model performance is regained with retraining.
>
> "Analysis on this would be nice to see..."
> You are correct that it’s not the exact same number (see Figure 5) but we are careful to make sure that the final test-accuracy reported is the same as the baseline or better by a small amount (e.g., VGG16 experiences an absolute improvement of less than 0.1% overall test accuracy).
> Figure 7 further shows how Weightless offers better compression vs. error scaling than CSR.
>
> 2) "Resutls are focused on fully connected layers which carry (for the given models)..."
> The models we chose were done so as they are the ones most commonly used in the literature (Deep Compression, DNS, and HashedNets). We also consider CNNs and show that Bloomier filters perform well on them (see LeNet5). Our findings suggest that so long as weights exhibit sufficient sparsity, the method is effective.
>
> As future work, we are actively looking into more advanced pruning techniques to achieve the necessary sparsity to encode networks like ResNet-34. We recently evaluated magnitude pruning on ResNet-34, but saw substantial increase in model error which we felt would be an unfair comparison.
>
> 3) "I would like to see further analysis on why Bloomier filter encoding improves accuracy (or is a typo and meant to be error?) by 2%..."
> You are correct in that this is a typo. It should be error and this is corrected.
>
> 4) "It is interesting to me how the retraining process is 'hidden' all over the paper..."
> We have now included a plot (Figure 5) which shows how retraining recovers accuracy in encoded layers. We have also included numbers for construction and reconstruction for all the largest layers in the models used (at the request of another reviewer). We find that on a modern machine, the longest construction takes is 8.5 minutes; the machines we used originally were older and part of a cluster being used by others.
>
> 5) "Interestingly, as mentioned in the paper, this is 'static compression'..."
> That is correct. We are looking into ways to compute in the compressed space with Weightless, but to be competitive it will likely require special hardware and require a deeper investigation.
>
> We did not intend to mislead the reader, this is a compression paper for efficient weight transmission (and storage). If there is a way we could fix this we will gladly amend the paper.
>
> 6) "I would like to see the sensibility analysis with respect to t and the number of clusters."
> We have added a plot (Figure 6) to show this to the appendix.
>
> 7) "As mentioned before, LeNet is great but would be nice to see more complicated models (even resnet on CIFAR)..."
> See above.
>
> 8) "Results are focused on a single layer. What happens if all the layers are considered at the same time?..."
> If all the layers are considered (i.e., encoded) at the same time, there is no opportunity for deeper layers to be retrained to compensate for errors in the earlier layers. In this scenario, one would likely have to increase the t value to mitigate false positives or incur a slight increase in model error.
>
> If each layer is encoded individually, the process occurs precisely as specified in the paper. Each layer is encoded and the deeper layers are retrained around their false positives.

---

### Decision · Program_Chairs · 2018-01-29
**ICLR 2018 Conference Acceptance Decision**

**Decision:**

Invite to Workshop Track

**Comment:**

Pros:
-- Use of Bloomier filters for lossy compression of nets is novel and well motivated, with interesting compression performance.
Cons:
-- Does lossy compression for transmission, doesn’t address FLOPS required for runtime execution. A lot of times, client devices do not have enough cpu to run large networks (title should be udpated to mean compression and transmission)
-- Missing results for full network, larger deeper network.

Overall, the content is novel and interesting, so I would encourage the authors to submit to the workshop track.